# Epidemiological hypothesis testing using a phylogeographic and phylodynamic framework

Simon Dellicour [1,2 ✉], Sebastian Lequime [2], Bram Vrancken [2], Mandev S. Gill[2], Paul Bastide [2], Karthik Gangavarapu[3], Nathaniel L. Matteson[3], Yi Tan[4,5], Louis du Plessis [6], Alexander A. Fisher[7], Martha I. Nelson[8], Marius Gilbert[1], Marc A. Suchard [7,9,10], Kristian G. Andersen[3,11], Nathan D. Grubaugh[12], Oliver G. Pybus [6] & Philippe Lemey [2]

Computational analyses of pathogen genomes are increasingly used to unravel the dispersal history and transmission dynamics of epidemics. Here, we show how to go beyond historical reconstructions and use spatially-explicit phylogeographic and phylodynamic approaches to formally test epidemiological hypotheses. We illustrate our approach by focusing on the West Nile virus (WNV) spread in North America that has substantially impacted public, veterinary, and wildlife health. We apply an analytical workflow to a comprehensive WNV genome collection to test the impact of environmental factors on the dispersal of viral lineages and on viral population genetic diversity through time. We find that WNV lineages tend to disperse faster in areas with higher temperatures and we identify temporal variation in temperature as a main predictor of viral genetic diversity through time. By contrasting inference with simulation, we find no evidence for viral lineages to preferentially circulate within the same migratory bird flyway, suggesting a substantial role for non-migratory birds or mosquito dispersal along the longitudinal gradient.

[1] Spatial Epidemiology Lab (SpELL), Université Libre de Bruxelles, CP160/12, 50 Avenue FD Roosevelt, 1050 Bruxelles, Belgium. [2] Department of Microbiology, Immunology and Transplantation, Rega Institute, KU Leuven, Herestraat 49, 3000 Leuven, Belgium. [3] Department of Immunology and Microbiology, The Scripps Research Institute, La Jolla, CA 92037, USA. [4] Department of Medicine, Vanderbilt University Medical Center, Nashville, TN, USA. [5] Infectious Diseases Group, J. Craig Venter Institute, Rockville, MD, USA. [6] Department of Zoology, University of Oxford, Oxford, UK. [7] Department of Biomathematics, David Geffen School of Medicine, University of California, Los Angeles, CA, USA. [8] Fogarty International Center, National Institutes of Health, Bethesda, MD 20894, USA. [9] Department of Biostatistics, Fielding School of Public Health, University of California, Los Angeles, CA, USA. [10] Department of Human Genetics, David Geffen School of Medicine, University of California, Los Angeles, CA, USA. [11] Scripps Research Translational Institute, La Jolla, CA 92037, USA. [12] Department of Epidemiology of Microbial Diseases, Yale School of Public Health, New Haven, CT 06510, USA. ✉email: simon.dellicour@ulb.ac.be

The evolutionary analysis of rapidly evolving pathogens, particularly RNA viruses, allows us to establish the epidemiological relatedness of cases through time and space. Such transmission information can be difficult to uncover using classical epidemiological approaches. The development of spatially explicit phylogeographic models[1,2], which place time-referenced phylogenies in a geographical context, can provide a detailed spatio-temporal picture of the dispersal history of viral lineages[3]. These spatially explicit reconstructions frequently serve illustrative or descriptive purposes, and remain underused for testing epidemiological hypotheses in a quantitative fashion. However, recent advances in methodology offer the ability to analyse the impact of underlying factors on the dispersal dynamics of virus lineages[4–6], giving rise to the concept of landscape phylogeography[7]. Similar improvements have been made to phylodynamic analyses that use flexible coalescent models to reconstruct virus demographic history[8,9]; these methods can now provide insights into epidemiological or environmental variables that might be associated with population size change[10].

In this study, we focus on the spread of West Nile virus (WNV) across North America, which has considerably impacted public, veterinary, and wildlife health[11]. WNV is the most widely distributed encephalitic flavivirus transmitted by the bite of infected mosquitoes[12,13]. WNV is a single-stranded RNA virus that is maintained by an enzootic transmission cycle primarily involving *Culex* mosquitoes and birds[14–17]. Humans are incidental terminal hosts, because viremia does not reach a sufficient level for subsequent transmission to mosquitoes[17,18]. WNV human infections are mostly subclinical although symptoms may range from fever to meningoencephalitis and can occasionally lead to death[17,19]. It has been estimated that only 20–25% of infected people become symptomatic, and that <1 in 150 develops neuroinvasive disease[20]. The WNV epidemic in North America likely resulted from a single introduction to the continent 20 years ago[21]. Its persistence is likely not the result of successive reintroductions from outside of the hemisphere, but rather of local overwintering and maintenance of long-term avian and/or mosquito transmission cycles[11]. Overwintering could also be facilitated by vertical transmission of WNV from infected female mosquitos to their offspring[22–24]. WNV represents one of the most important vector-borne diseases in North America[15]; there were an estimated 7 million human infections in the U.S.[25], causing a reported 24,657 human neuroinvasive cases between 1999 to 2018, leading to 2,199 deaths (www.cdc.gov/westnile). In addition, WNV has had a notable impact on North American bird populations[26,27], with several species[28] such as the American crow (*Corvus brachyrhynchos*) being particularly severely affected.

Since the beginning of the epidemic in North America in 1999[21], WNV has received considerable attention from local and national health institutions and the scientific community. This had led to the sequencing of >2000 complete viral genomes collected at various times and locations across the continent. The resulting availability of virus genetic data represents a unique opportunity to better understand the evolutionary history of WNV invasion into an originally non-endemic area. Here, we take advantage of these genomic data to address epidemiological questions that are challenging to tackle with non-molecular approaches.

The overall goal of this study is to go beyond historical reconstructions and formally test epidemiological hypotheses by exploiting phylodynamic and spatially explicit phylogeographic models. We detail and apply an analytical workflow that consists of state-of-the-art methods that we further improve to test hypotheses in molecular epidemiology. We demonstrate the power of this approach by analysing a comprehensive data set of WNV genomes with the objective of unveiling the dispersal and demographic dynamics of the virus in North America. Specifically, we aim to (i) reconstruct the dispersal history of WNV on the continent, (ii) compare the dispersal dynamics of the three WNV genotypes, (iii) test the impact of environmental factors on the dispersal locations of WNV lineages, (iv) test the impact of environmental factors on the dispersal velocity of WNV lineages, (v) test the impact of migratory bird flyways on the dispersal history of WNV lineages, and (vi) test the impact of environmental factors on viral genetic diversity through time.

## Results

**Reconstructing the dispersal history and dynamics of WNV lineages.** To infer the dispersal history of WNV lineages in North America, we performed a spatially explicit phylogeographic analysis[1] of 801 viral genomes (Supplementary Figs. S1 and S2), which is almost an order of magnitude larger than the early US-wide study by Pybus et al.[2] (104 WNV genomes). The resulting sampling presents a reasonable correspondence between West Nile fever prevalence in the human population and sampling density in most areas associated with the highest numbers of reported cases (e.g., Los Angeles, Houston, Dallas, Chicago, New York), but also some under-sampled locations (e.g., in Colorado; Supplementary Fig. S1). Year-by-year visualisation of the reconstructed invasion history highlights both relatively fast and relatively slow long-distance dispersal events across the continent (Supplementary Fig. S3), which is further confirmed by the comparison between the durations and geographic distances travelled by phylogeographic branches (Supplementary Fig. S4). Some of these long-distance dispersal events were notably fast, with >2000 km travelled in only a couple of months (Supplementary Fig. S4).

To quantify the spatial dissemination of virus lineages, we extracted the spatio-temporal information embedded in molecular clock phylogenies sampled by Bayesian phylogeographic analysis. From the resulting collection of lineage movement vectors, we estimated several key statistics of spatial dynamics (Fig. 1). We estimated a mean lineage dispersal velocity of ~1200 km/year, which is consistent with previous estimates[2]. We further inferred how the mean lineage dispersal velocity changed through time, and found that dispersal velocity was notably higher in the earlier years of the epidemic (Fig. 1). The early peak of lineage dispersal velocity around 2001 corresponds to the expansion phase of the epidemic. This is corroborated by our estimate of the maximal wavefront distance from the epidemic origin through time (Fig. 1). This expansion phase lasted until 2002, when WNV lineages first reached the west coast (Fig. 1 and Supplementary Fig. S3). From East to West, WNV lineages dispersed across various North American environmental conditions in terms of land cover, altitude, and climatic conditions (Fig. 2).

We also compared the dispersal velocity estimated for five subsets of WNV phylogenetic branches (Fig. 3): branches occurring during (before 2002) and after the expansion phase (after 2002), as well as branches assigned to each of the three commonly defined WNV genotypes that circulated in North America (NY99, WN02, and SW03; Supplementary Figs. S1–S2). While NY99 is the WNV genotype that initially invaded North America, WN02 and subsequently SW03 emerged as the two main co-circulating genotypes characterised by distinct amino acid substitutions[29–32]. We specifically compare the dispersal history and dynamics of lineages belonging to these three different genotypes in order to investigate the assumption that WNV dispersal might have been facilitated by local environmental adaptations[32]. To address this question, we performed the three landscape phylogeographic testing approaches presented

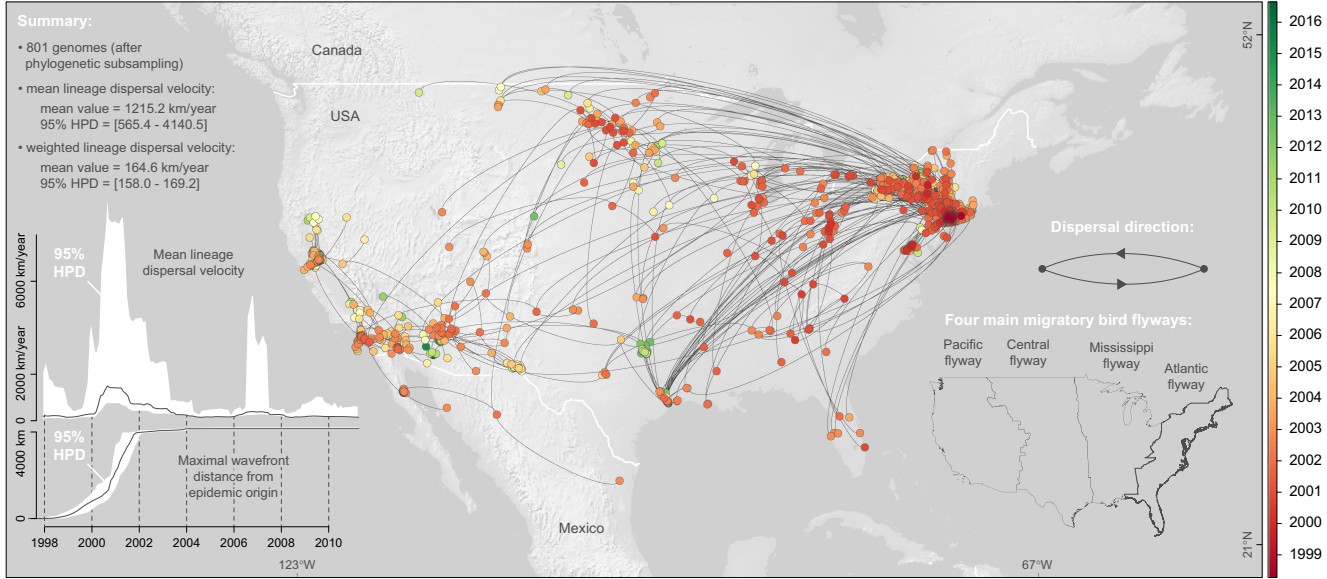

**Fig. 1 Spatio-temporal diffusion of WNV lineages in North America.** Maximum clade credibility (MCC) tree obtained by continuous phylogeographic inference based on 100 posterior trees (see the text for further details). Nodes of the tree are coloured from red (the time to the most recent common ancestor, TMRCA) to green (most recent sampling time). Older nodes are plotted on top of younger nodes, but we provide also an alternative year-by-year representation in Supplementary Fig. S1. In addition, this figure reports global dispersal statistics (mean lineage dispersal velocity and mean diffusion coefficient) averaged over the entire virus spread, the evolution of the mean lineage dispersal velocity through time, the evolution of the maximal wavefront distance from the origin of the epidemic, as well as the delimitations of the North American Migratory Flyways (NAMF) considered in the USA.

**Fig. 2 Environmental variables tested for their impact on the dispersal of West Nile virus lineages in North America.** See Table S1 for the source of data for each environmental raster.

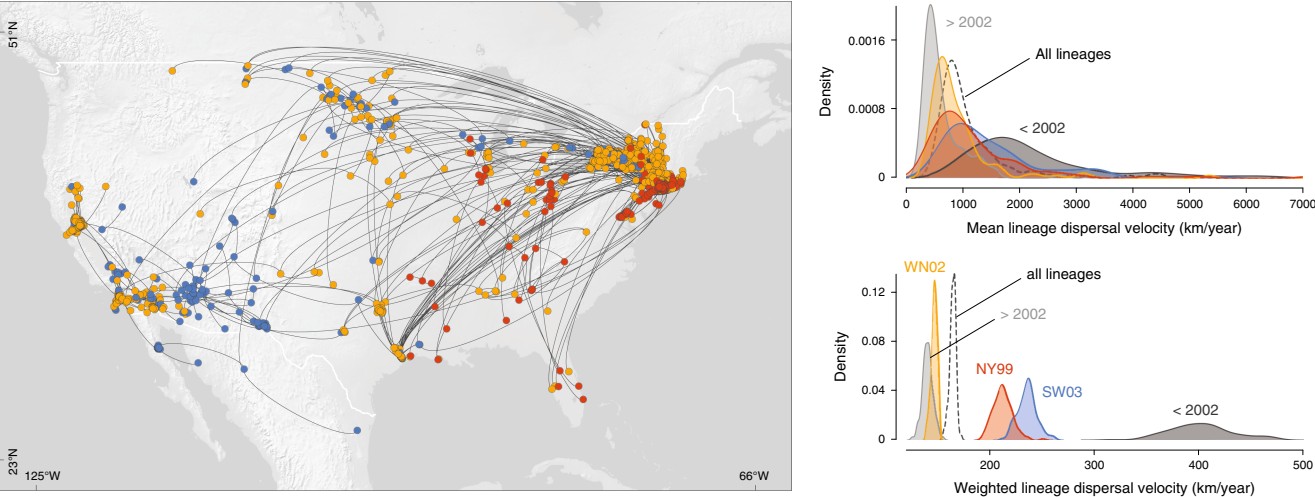

**Fig. 3 Comparison of the dispersal history and velocity of WNV lineages belonging to three phenotypically relevant genotypes (NY99, WN02, and SW03).** The map displays the maximum clade credibility (MCC) tree obtained by continuous phylogeographic inference with nodes coloured according to three different genotypes.

below on the complete data set including all viral lineages inferred by continuous phylogeographic inference, as well as on these different subsets of lineages. We first compared the lineage dispersal velocity estimated for each subset by estimating both the mean and weighted lineage dispersal velocities. As shown in Fig. 3 and detailed in the Methods section, the weighted metric is more efficient and suitable to compare the dispersal velocity associated with different data sets, or subsets in the present situation. Posterior distributions of the weighted dispersal velocity confirmed that the lineage dispersal was much faster during the expansion phase (<2002; Fig. 3). Second, these estimates also indicated that SW03 is associated with a higher dispersal velocity than the dominant genotype, WN02.

**Testing the impact of environmental factors on the dispersal locations of viral lineages.** To investigate the impact of environmental factors on the dispersal dynamics of WNV, we performed three different statistical tests in a landscape phylogeographic framework. First, we tested whether lineage dispersal locations tended to be associated with specific environmental conditions. In practice, we started by computing the $E$ statistic, which measures the mean environmental values at tree node positions. These values were extracted from rasters (geo-referenced grids) that summarised the different environmental factors to be tested: elevation, main land cover variables in the study area (forests, shrublands, savannas, grasslands, croplands, urban areas; Fig. 2), and monthly time-series collections of climatic rasters (for temperature and precipitation; Supplementary Table S1). For the time-series climatic factors, the raster used for extracting the environmental value was selected according to the time of occurrence of each tree node. The $E$ statistic was computed for each posterior tree sampled during the phylogeographic analysis, yielding a posterior distribution of this metric (Supplementary Fig. S5). To determine whether the posterior distributions for $E$ were significantly lower or higher than expected by chance under a null dispersal model, we also computed $E$ based on simulations, using the inferred set of tree topologies along which a new stochastic diffusion history was simulated according to the estimated diffusion parameters. The statistical support was assessed by comparing inferred and simulated distributions of $E$. If the inferred distribution was significantly lower than the simulated distribution of $E$, this provides evidence for the environmental factor to repulse viral lineages, while an inferred

distribution higher than the simulated distribution of $E$ would provide evidence for the environmental factor to attract viral lineages.

These first landscape phylogeographic tests revealed that WNV lineages (i) tended to avoid areas associated with relatively higher elevation, forest coverage, and precipitation, and (ii) tended to disperse in areas associated with relatively higher urban coverage, temperature, and shrublands (Supplementary Table S2). However, when analysing each genotype separately, different trends emerged. For instance, SW03 lineages did not tend to significantly avoid (or disperse to) areas with relatively higher elevation (~600–750 m above sea level), and only SW03 lineages significantly dispersed towards areas with shrublands (Supplementary Table S2). Furthermore, when only focusing on WNV lineages occurring before 2002, we did not identify any significant association between environmental values and node positions. Interestingly, this implies that we cannot associate viral dispersal during the expansion phase with specific favourable environmental conditions (Supplementary Table S2). As these tests are directly based on the environmental values extracted at internal and tip node positions, their outcome can be particularly impacted by the nature of sampling. Indeed, half of the node positions, i.e., the tip node positions, are directly determined by the sampling. To assess the sensitivity of the tests to heterogeneous sampling, we also repeated these tests while only considering internal tree nodes. Since internal nodes are phylogeographically linked to tip nodes, discarding tip branches can only mitigate the direct impact of the sampling pattern on the outcome of the analysis. These additional tests provided consistent results, except for one environmental factor: while precipitation was identified as a factor repulsing viral lineages, it was not the case anymore when only considering internal tree branches, indicating that the initial result could be attributed to a sampling artefact.

**Testing the impact of environmental factors on the dispersal velocity of viral lineages.** In the second landscape phylogeographic test, we analysed whether the heterogeneity observed in lineage dispersal velocity could be explained by specific environmental factors that are predominant in the study area. For this purpose, we used a computational method that assesses the correlation between lineage dispersal durations and environmentally scaled distances[4,33]. These distances were computed on several environmental rasters (Fig. 2 and Supplementary Table S1):

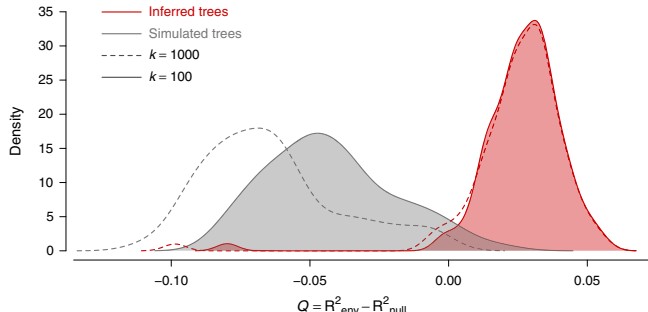

**Fig. 4 Impact of annual mean temperature acting as a conductance factor on the dispersal velocity of viral lineages.** The graph displays the distribution of the correlation metric $Q$ computed on 100 spatially annotated trees obtained by continuous phylogeographic inference (red distributions). The metric $Q$ measures to what extent considering a heterogeneous environmental raster, increases the correlation between lineage durations and environmentally scaled distances compared to a homogeneous raster. If $Q$ is positive and supported, it indicates that the heterogeneity in lineage dispersal velocity can be at least partially explained by the environmental factor under investigation. The graph also displays the distribution of $Q$ values computed on the same 100 posterior trees along which we simulated a new forward-in-time diffusion process (grey distributions). These simulations are used as a null dispersal model to estimate the support associated with the inferred distribution of $Q$ values. For both inferred and simulated trees, we report the $Q$ distributions obtained while transforming the original environmental raster according to two different scaling parameter $k$ values (100 and 1000; respectively full and dashed line, see the text for further details on this transformation). The annual mean temperature raster, transformed in conductance values using these two $k$ values, is the only environmental factor for which we detect a positive distribution of $Q$ that is also associated with a strong statistical support (Bayes factor > 20).

elevation, main land cover variables in the study area (forests, shrublands, savannas, grasslands, croplands, urban areas), as well as annual mean temperature and annual precipitation. This analysis aimed to quantify the impact of each factor on virus movement by calculating a statistic, $Q$, that measures the correlation between lineage durations and environmentally scaled distances. Specifically, the $Q$ statistic describes the difference in strength of the correlation when distances are scaled using the environmental raster versus when they are computed using a "null" raster (i.e., a uniform raster with a value of "1" assigned to all cells). As detailed in the Methods section, two alternative path models were used to compute these environmentally scaled distances: the least-cost path model[34] and a model based on circuit theory[35]. The $Q$ statistic was estimated for each posterior tree sampled during the phylogeographic analysis, yielding a posterior distribution of this metric. As for the statistic $E$, statistical support for $Q$ was then obtained by comparing inferred and simulated distributions of $Q$; the latter was obtained by estimating $Q$ on the same set of tree topologies, along which a new stochastic diffusion history was simulated. This simulation procedure thereby generated a null model of dispersal, and the comparison between the inferred and simulated $Q$ distributions enabled us to approximate a Bayes factor support (see "Methods" for further details).

As summarised in Supplementary Table S3, we found strong support for one variable: annual temperature raster treated as a conductance factor. Using this factor, the association between lineage duration and environmentally scaled distances was significant using the path model based on circuit theory[35]. As detailed in Fig. 4, this environmental variable better explained the heterogeneity in lineage dispersal velocity than geographic distance alone (i.e., its $Q$ distribution was positive). Furthermore,

this result received strong statistical support (Bayes factor > 20), obtained by comparing the distribution of $Q$ values with that obtained under a null model (Fig. 4). We also performed these tests on each WNV genotype separately (Supplementary Table S4). With these additional tests, we only found the same statistical support associated with temperature for the viral lineages belonging to the WN02 genotype. In addition, these tests based on subsets of lineages also revealed that the higher elevation was significantly associated with lower dispersal velocity of WN02 lineages.

**Testing the impact of environmental factors on the dispersal frequency of viral lineages.** The third landscape phylogeography test that we performed focused on the impact of specific environmental factors on the dispersal frequency of viral lineages. Specifically, we aimed to investigate the impact of migratory bird flyways on the dispersal history of WNV. For this purpose, we first tested whether virus lineages tended to remain within the same North American Migratory Flyway (NAMF; Fig. 1). As in the two first testing approaches, we again compared inferred and simulated diffusion dynamics (i.e., simulation of a new stochastic diffusion process along the estimated trees). Under the null hypothesis (i.e., NAMFs have no impact on WNV dispersal history), virus lineages should not transition between flyways less often than under the null dispersal model. Our test did not reject this null hypothesis (BF < 1). As the NAMF borders are based on administrative areas (US counties), we also performed a similar test using the alternative delimitation of migratory bird flyways estimated for terrestrial bird species by La Sorte et al.[36] (Supplementary Fig. S6). Again, the null hypothesis was not rejected, indicating that inferred virus lineages did not tend to remain within specific flyways more often than expected by chance. Finally, these tests were repeated on each of the five subsets of WNV lineages (<2002, >2002, NY99, WN02, SW03) and yielded the same results, i.e., no rejection of the null hypothesis stating that flyways do not constrain WNV dispersal.

**Testing the impact of environmental factors on the viral genetic diversity through time.** We next employed a phylodynamic approach to investigate predictors of the dynamics of viral genetic diversity through time. In particular, we used the generalised linear model (GLM) extension[10] of the skygrid coalescent model[9], hereafter referred to as the "skygrid-GLM" approach, to statistically test for associations between estimated dynamics of virus effective population size and several covariates. Coalescent models that estimate effective population size (Ne) typically assume a single panmictic population that encompasses all individuals. As this assumption is frequently violated in practice, the estimated effective population size is sometimes interpreted as representing an estimate of the genetic diversity of the whole virus population[37]. The skygrid-GLM approach accounts for uncertainty in effective population size estimates when testing for associations with covariates; neglecting this uncertainty can lead to spurious conclusions[10].

We first performed univariate skygrid-GLM analyses of four distinct time-varying covariates reflecting seasonal changes: monthly human WNV case counts (log-transformed), temperature, precipitation, and a greenness index. For the human case count covariate, we only detected a significant association with the viral effective population size when considering a lag period of at least one month. In addition, univariate analyses of temperature and precipitation time-series were also associated with the virus genetic diversity dynamics (i.e., the posterior GLM coefficients for these covariates had 95% credible intervals that did not include zero; Fig. 5). To further assess the relative

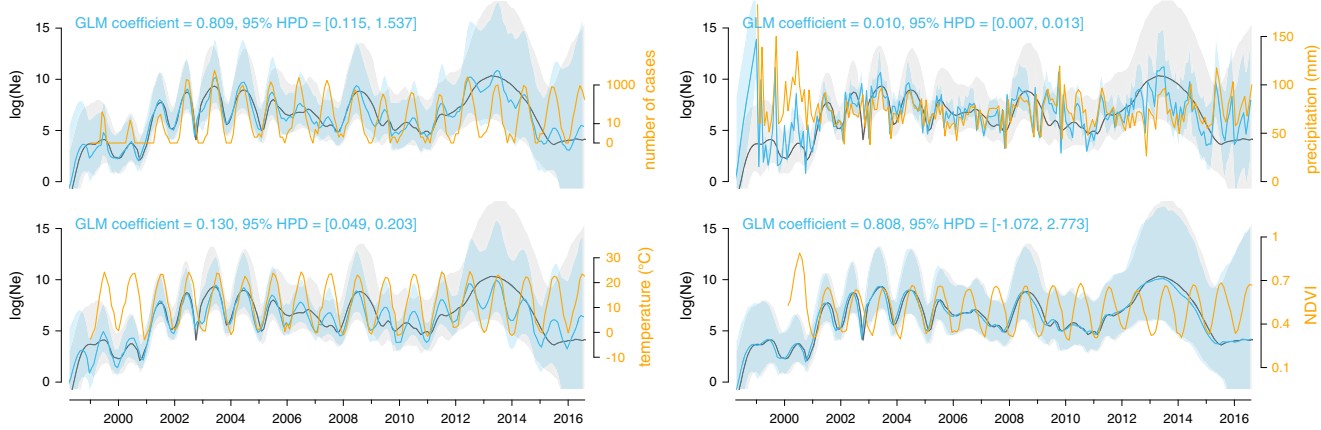

**Fig. 5 Associations between viral effective population size and potential covariates.** These associations were tested with a generalised linear model (GLM) extension of the coalescent model used to infer the dynamics of the viral effective population size of the virus (Ne) through time. Specifically, we here tested the following time-series variables as potential covariates (orange curves): number of human cases (log-transformed and with a negative time period of one month), mean temperature, mean precipitation, and Normalised Difference Vegetation Index (NDVI, a greenness index). Posterior mean estimates of the viral effective population size based on both sequence data and covariate data are represented by blue curves, and the corresponding blue polygon reflects the 95% HPD region. Posterior mean estimates of the viral effective population size inferred strictly from sequence data are represented by grey curves and the corresponding grey polygon reflects the 95% HPD region. A significant association between the covariate and effective population size is inferred when the 95% HPD interval of the GLM coefficient excludes zero, which is the case for the case count, temperature, and precipitation covariates.

importance of each covariate, we performed multivariate skygrid-GLM analyses to rank covariates based on their inclusion probabilities[38]. The first multivariate analysis involved all covariates and suggested that the lagged human case counts best explain viral population size dynamics, with an inclusion probability close to 1. However, because human case counts are known to be a consequence rather than a potential causal driver of the WNV epidemic, we performed a second multivariate analysis after having excluded this covariate. This time, the temperature time-series emerged as the covariate with the highest inclusion probability.

## Discussion

In this study, we use spatially explicit phylogeographic and phylodynamic inference to reconstruct the dispersal history and dynamics of a continental viral spread. Through comparative analyses of lineage dispersal statistics, we highlight distinct trends within the overall spread of WNV. First, we have demonstrated that the WNV spread in North America can be divided into an initial "invasion phase" and a subsequent "maintenance phase" (see Carrington et al.[39] for similar terminology used in the context of spatial invasion of dengue viruses). The invasion phase is characterised by an increase in virus effective population size until the west coast was reached, followed by a maintenance phase associated with a more stable cyclic variation of effective population size (Fig. 5). In only 2–3 years, WNV rapidly spread from the east to the west coast of North America, despite the fact that the migratory flyways of its avian hosts are primarily north-south directed. This could suggest potentially important roles for non-migratory bird movements, as well as natural or human-mediated mosquito dispersal, in spreading WNV along a longitudinal gradient[40,41]. However, the absence of clear within flyway clustering of viral lineages could also arise when different avian migration routes intersect at southern connections. If local WNV transmission occurs at these locations, viruses could travel along different flyways when the birds make their return northward migration, as proposed by Swetnam et al.[42]. While this scenario is possible, there is insufficient data to formally investigate with our approaches. Overall, we uncover a higher lineage dispersal velocity during the invasion phase, which could reflect a

consequence of increased bird immunity through time slowing down spatial dispersal. It has indeed been demonstrated that avian immunity can impact WNV transmission dynamics[43]. Second, we also reveal different dispersal velocities associated with the three WNV genotypes that have circulated in North America: viral lineages of the dominant current genotype (WN02) have spread slower than lineages of NY99 and SW03. NY99 was the main genotype during the invasion phase but has not been detected in the US since the mid 2000s. A faster dispersal associated with NY99 is thus coherent with the higher dispersal velocity identified for lineages circulating during the invasion phase. The higher dispersal velocity for SW03 compared to WN02 is in line with recently reported evidence that SW03 spread faster than WN02 in California[44].

In the second part of the study, we illustrate the application of a phylogeographic framework for hypothesis testing that builds on previously developed models. These analytical approaches are based on a spatially explicit phylogeographic or phylodynamic (skygrid coalescent) reconstruction, and aim to assess the impact of environmental factors on the dispersal locations, velocity, and frequency of viral lineages, as well as on the overall genetic diversity of the viral population. The WNV epidemic in North America is a powerful illustration of viral invasion and emergence in a new environment[31], making it a highly relevant case study to apply such hypothesis testing approaches. We first test the association between environmental factors and lineage dispersal locations, demonstrating that, overall, WNV lineages have preferentially circulated in specific environmental conditions (higher urban coverage, temperature, and shrublands) and tended to avoid others (higher elevation and forest coverage). Second, we have tested the association between environmental factors and lineage dispersal velocity. With these tests, we find evidence for the impact of only one environmental factor on virus lineage dispersal velocity, namely annual mean temperature. Third, we tested the impact of migratory flyways on the dispersal frequency of viral lineages among areas. Here, we formally test the hypothesis that WNV lineages are contained or preferentially circulate within the same migratory flyway and find no statistical support for this.

We have also performed these three different landscape phylogeographic tests on subsets of WNV lineages (lineages occurring during and after the invasion phase, as well as NY99, WN02, and SW03 lineages). When focusing on lineages occurring during the invasion phase (<2002), we do not identify any significant association between a particular environmental factor and the dispersal location or velocity of lineages. This result corroborates the idea that, during the early phase of the epidemic, the virus rapidly conquered the continent despite various environmental conditions, which was likely helped by large populations of susceptible hosts/vectors already present in North America[32]. These additional tests also highlight interesting differences among the three WNV genotypes. For instance, we found that the dispersal of SW03 genotype is faster than WN02 and also preferentially in shrublands and at higher temperatures. At face value, it appears that the mutations that define the SW03 genotype, NS4A-A85T and NS5-K314R[45], may be signatures of adaptations to such specific environmental conditions. It may, however, be an artefact of the SW03 genotype being most commonly detected in mosquito species such as *Cx. tarsalis* and *Cx. quiquefasciatus* that are found in the relatively high elevation shrublands of the southwest US[44,46]. In this scenario, the faster dispersal velocities could result from preferentially utilising these two highly efficient WNV vectors[47], especially when considering the warm temperatures of the southwest[48,49]. It is also important to note that to date, no specific phenotypic advantage has been observed for SW03 genotypes compared to WN02 genotypes[50,51]. Further research is needed to discern if the differences among the three WNV genotypes are due to virus-specific factors, heterogeneous sampling effort, or ecological variation.

When testing the impact of flyways on the five different subsets of lineages, we reach the same result of no preferential circulation within flyways. This overall result contrasts with previously reported phylogenetic clustering by flyways[31,42]. However, the clustering analysis of Di Giallonardo et al.[31] was based on a discrete phylogeographic analysis and, as recognised by the authors, it is difficult to distinguish the effect of these flyways from those of geographic distance. Here, we circumvent this issue by performing a spatial analysis that explicitly represents dispersal as a function of geographic distance. Our results are, however, not in contradiction with the already established role of migratory birds in spreading the virus[52,53], but we do not find evidence that viral lineage dispersal is structured by flyway. Specifically, our test does not reject the null hypothesis of absence of clustering by flyways, which at least signals that the tested flyways do not have a discernible impact on WNV lineages circulation. Dissecting the precise involvement of migratory bird in WNV spread, thus, require additional collection of empirical data. Furthermore, our phylogeographic analysis highlights the occurrence of several fast and long-distance dispersal events along a longitudinal gradient. A potential anthropogenic contribution to such long-distance dispersal (e.g., through commercial transport) warrants further investigation.

In addition to its significant association with the dispersal locations and velocity of WNV lineages, the relevance of temperature is further demonstrated by the association between the virus genetic dynamics and several time-dependent covariates. Indeed, among the three environmental time-series we tested, temporal variation in temperature is the most important predictor of cycles in viral genetic diversity. Temperature is known to have a dramatic impact on the biology of arboviruses and their arthropod hosts[54], including WNV. Higher temperatures have been shown to impact directly the mosquito life cycle, by accelerating larval development[11], decreasing the interval between blood meals, and prolonging the mosquito breeding season[55]. Higher temperatures have been also associated with shorter

extrinsic incubation periods, accelerating WNV transmission by the mosquito vector[56,57]. Interestingly, temperature has also been suggested as a variable that can increase the predictive power of WNV forecast models[58]. The impact of temperature that we reveal here on both dispersal velocity and viral genetic diversity is particularly important in the context of global warming. In addition to altering mosquito species distribution[59,60], an overall temperature increase in North America could imply an increase in enzootic transmission and hence increased spill-over risk in different regions. In addition to temperature, we find evidence for an association between viral genetic diversity dynamics and the number of human cases, but only when a lag period of at least one month is added to the model (having only monthly case counts available, it was not possible to test shorter lag periods). Such lag could, at least in part, be explained by the time needed for mosquitos to become infectious and bite humans. As human case counts are in theory backdated to the date of onset of illness, incubation time in humans should not contribute to this lag.

Our study illustrates and details the utility of landscape phylogeographic and phylodynamic hypothesis tests when applied to a comprehensive data set of viral genomes sampled during an epidemic. Such spatially explicit investigations are only possible when viral genomes (whether recently collected or available on public databases such as GenBank) are associated with sufficiently precise metadata, in particular the collection date and the sampling location. The availability of precise collection dates - ideally known to the day - for isolates obtained over a sufficiently long time-span enables reliable timing of epidemic events due to the accurate calibration of molecular clock models. Further, spatially explicit phylogeographic inference is possible only when viral genomes are associated with sampling coordinates. However, geographic coordinates are frequently unknown or unreported. In practice this may not represent a limitation if a sufficiently precise descriptive sampling location is specified (e.g., a district or administrative area), as this information can be converted into geographic coordinates. The full benefits of comprehensive phylogeographic analyses of viral epidemics will be realised only when precise location and time metadata are made systematically available.

Although we use a comprehensive collection of WNV genomes in this study, it would be useful to perform analyses based on even larger data sets that cover regions under-sampled in the current study; this work is the focus of an ongoing collaborative project (westnile4k.org). While the resolution of phylogeographic analyses will always depend on the spatial granularity of available samples, they can still be powerful in elucidating the dispersal history of sampled lineages. When testing the impact of environmental factors on lineage dispersal velocity and frequency, heterogeneous sampling density will primarily affect statistical power in detecting the impact of relevant environmental factors in under- or unsampled areas[33]. However, the sampling pattern can have much more impact on the tests dedicated to the impact of environmental factors on the dispersal locations of viral lineages. As stated above, in this test, half of the environmental values will be extracted at tip node locations, which are directly determined by the sampling effort. To circumvent this issue and assess the robustness of the test regarding the sampling pattern, we here proposed to repeat the analysis after having discarded all the tip branches, which logically mitigated a potential impact of the sampling pattern on the outcome of this analysis. Furthermore, in this study, we note that heterogeneous sampling density across counties can be at least partially mitigated by performing phylogenetic subsampling (detailed in the "Methods" section). Another limitation to underline is that, contrary to the tests focusing on the impact of environmental factors on the dispersal locations and frequency, the present framework does not allow

testing the impact of time-series environmental variables on the dispersal velocity of viral lineages. It would be interesting to extend that framework so that it can, e.g., test the impact of spatio-temporal variation of temperature on the dispersal velocity of WNV lineages. On the opposite, while skygrid-GLM analyses intrinsically integrate temporal variations of covariates, these tests treat the epidemic as a unique panmictic population of viruses. In addition to ignoring the actual population structure, this aspect implies the comparison of the viral effective population size with a unique environmental value per time slice and for the entire study area. To mitigate spatial heterogeneity as much as possible, we used the continuous phylogeographic reconstruction to define successive minimum convex hull polygons delimiting the study area at each time slice. These polygons were used to extract the environmental values that were then averaged to obtain a single environmental value per time slice considered in the skygrid-GLM analysis.

By placing virus lineages in a spatio-temporal context, phylogeographic inference provides information on the linkage of infections through space and time. Mapping lineage dispersal can provide a valuable source of information for epidemiological investigations and can shed light on the ecological and environmental processes that have impacted the epidemic dispersal history and transmission dynamics. When complemented with phylodynamic testing approaches, such as the skygrid-GLM approach used here, these methods offer new opportunities for epidemiological hypotheses testing. These tests can complement traditional epidemiological approaches that employ occurrence data. If coupled to real-time virus genome sequencing, landscape phylogeographic and phylodynamic testing approaches have the potential to inform epidemic control and surveillance decisions[61].

## Methods

**Selection of viral sequences**. We started by gathering all WNV sequences available on GenBank on the 20th November 2017. We only selected sequences (i) of at least 10 kb, i.e., covering almost the entire viral genome (~11 kb), and (ii) associated with a sufficiently precise sampling location, i.e., at least an administrative area of level 2. Administrative areas of level 2 are hereafter abbreviated "admin-2" and correspond to US counties. Finding the most precise sampling location (admin-2, city, village, or geographic coordinates), as well as the most precise sampling date available for each sequence, required a bibliographic screening because such metadata are often missing on GenBank. The resulting alignment of 993 geo-referenced genomic sequences of at least 10 kb was made using MAFFT[62] and manually edited in AliView[63]. Based on this alignment, we performed a first phylogenetic analysis using the maximum likelihood method implemented in the programme FastTree[64] with 1000 bootstrap replicates to assess branch supports. The aim of this preliminary phylogenetic inference was solely to identify monophyletic clades of sequences sampled from the same admin-2 area associated with a bootstrap support higher than 70%. Such phylogenetic clusters of sampled sequences largely represent lineage dispersal within a specific admin-2 area. As we randomly draw geographic coordinates from an admin-2 polygon for sequences only associated with an admin-2 area of origin, keeping more than one sequence per phylogenetic cluster would not contribute any meaningful information in subsequent phylogeographic analyses[61]. Therefore, we subsampled the original alignment such that only one sequence is randomly selected per phylogenetic cluster, leading to a final alignment of 801 genomic sequences (Supplementary Fig. S1). In the end, selected sequences were mostly derived from mosquitoes (~50%) and birds (~44%), with very few (~5%) from humans.

**Time-scaled phylogenetic analysis**. Time-scaled phylogenetic trees were inferred using BEAST 1.10.4[65] and the BEAGLE 3 library[66] to improve computational performance. The substitution process was modelled according to a GTR+Γ parametrisation[67], branch-specific evolutionary rates were modelled according to a relaxed molecular clock with an underlying log-normal distribution[68], and the flexible skygrid model was specified as tree prior[9,10]. We ran and eventually combined ten independent analyses, sampling Markov chain Monte-Carlo (MCMC) chains every $2 \times 10^8$ generations. Combined, the different analyses were run for $>10^{12}$ generations. For each distinct analysis, the number of sampled trees to discard as burn-in was identified using Tracer 1.7[69]. We used Tracer to inspect the convergence and mixing properties of the combined output, referred to as the "skygrid analysis" throughout the text, to ensure that estimated sampling size (ESS) values associated with estimated parameters were all >200.

**Spatially explicit phylogeographic analysis**. The spatially explicit phylogeographic analysis was performed using the relaxed random walk (RRW) diffusion model implemented in BEAST[1,2]. This model allows the inference of spatially and temporally referenced phylogenies while accommodating variation in dispersal velocity among branches[3]. Following Pybus et al.[2], we used a gamma distribution to model the among-branch heterogeneity in diffusion velocity. Even when launching multiple analyses and using GPU resources to speed-up the analyses, poor MCMC mixing did not permit reaching an adequate sample from the posterior in a reasonable amount of time. This represents a challenging problem that is currently under further investigation[70]. To circumvent this issue, we performed 100 independent phylogeographic analyses each based on a distinct fixed tree sampled from the posterior distribution of the skygrid analysis. We ran each analysis until ESS values associated with estimated parameters were all greater than 100. We then extracted the last spatially annotated tree sampled in each of the 100 posterior distributions, which is the equivalent of randomly sampling a post-burn-in tree within each distribution. All the subsequent landscape phylogeographic testing approaches were based on the resulting distribution of the 100 spatially annotated trees. Given the computational limitations, we argue that the collection of 100 spatially annotated trees, extracted from distinct posterior distributions each based on a different fixed tree topology, represents a reasonable approach to obtain a phylogeographic reconstruction that accounts for phylogenetic uncertainty. We note that this is similar to the approach of using a set of empirical trees that is frequently employed for discrete phylogeographic inference[71,72], but direct integration over such a set of trees is not appropriate for the RRW model because the proposal distribution for branch-specific scaling factors does not hold in this case. We used TreeAnnotator 1.10.4[65] to obtain the maximum clade credibility (MCC) tree representation of the spatially explicit phylogeographic reconstruction (Supplementary Fig. S2).

In addition to the overall data set encompassing all lineages, we also considered five different subsets of lineages: phylogeny branches occurring before or after the end of the expansion/invasion phase (i.e., 2002; Fig. 1), as well as phylogeny branches assigned to each of the three WNV genotypes circulating in North America (NY99, WN02, and SW03; Supplementary Figs. S1–S2). These genotypes were identified on the basis of the WNV database published on the platform Nextstrain[32,73]. For the purpose of comparison, we performed all the subsequent landscape phylogeographic approaches on the overall data set but also on these five different subsets of WNV lineages.

**Estimating and comparing lineage dispersal statistics**. Phylogenetic branches, or "lineages", from spatially and temporally referenced trees can be treated as conditionally independent movement vectors[2]. We used the R package "seraphim"[74] to extract the spatio-temporal information embedded within each tree and to summarise lineages as movement vectors. We further used the package "seraphim" to estimate two dispersal statistics based on the collection of such vectors: the mean lineage dispersal velocity ($v_{mean}$) and the weighted lineage dispersal velocity ($v_{weighted}$)[74]. While both metrics measure the dispersal velocity associated with phylogeny branches, the second version involves a weighting by branch time[75]:

$$v_{mean} = \frac{1}{n}\sum_{i=1}^{n}\frac{d_i}{t_i} \quad \text{and} \quad v_{weighted} = \frac{\sum_{i=1}^{n}d_i}{\sum_{i=1}^{n}t_i}, \quad (1)$$

where $d_i$ and $t_i$ are the geographic distance travelled (great-circle distance in km) and the time elapsed (in years) on each phylogeny branch, respectively. The weighted metric is useful for comparing branch dispersal velocity between different data sets or different subsets of the same data set. Indeed, phylogeny branches with short duration have a lower impact on the weighted lineage dispersal velocity, which results in lower-variance estimates facilitating data set comparison[33]. On the other hand, estimating mean lineage dispersal velocity is useful when aiming to investigate the variability of lineage dispersal velocity within a distinct data set[75]. Finally, we also estimated the evolution of the maximal wavefront distance from the epidemic origin, as well as the evolution of the mean lineage dispersal velocity through time.

**Generating a null dispersal model of viral lineages dispersal**. To generate a null dispersal model we simulated a forward-in-time RRW diffusion process along each tree topology used for the phylogeographic analyses. These RRW simulations were performed with the "simulatorRRW1" function of the R package "seraphim" and based on the sampled precision matrix parameters estimated by the phylogeographic analyses[61]. For each tree, the RRW simulation started from the root node position inferred by the phylogeographic analysis. Furthermore, these simulations were constrained such that the simulated node positions remain within the study area, which is here defined by the minimum convex hull built around all node positions, minus non-accessible sea areas. As for the annotated trees obtained by phylogeographic inference, hereafter referred to as "inferred trees", we extracted the spatio-temporal information embedded within their simulated counterparts, hereafter referred as "simulated trees". As RRW diffusion processes were simulated along fixed tree topologies, each simulated tree shares a common topology with an inferred tree. Such a pair of inferred and simulated trees, thus, only differs by the geographic coordinates associated with their nodes, except for the root node position that was fixed as starting points for the RRW simulation. The distribution

of 100 simulated trees served as a null dispersal model for the landscape phylogeographic testing approaches described below.

**Testing the impact of environmental factors on the dispersal locations of viral lineages.** The first landscape phylogeographic testing approach consisted of testing the association between environmental conditions and dispersal locations of viral lineages. We started by simply visualising and comparing the environmental values explored by viral lineages. For each posterior tree sampled during the phylogeographic analysis, we extracted and then averaged the environmental values at the tree node positions. We then obtained, for each analysed environmental factor, a posterior distribution of mean environmental values at tree node positions for the overall data set as well as for the five subsets of WNV lineages described above. In addition to this visualisation, we also performed a formal test comparing mean environmental values extracted at node positions in inferred ($E_{estimated}$) and simulated trees ($E_{simulated}$). $E_{simulated}$ values constituted the distribution of mean environmental values explored under the null dispersal model, i.e., under a dispersal scenario that is not impacted by any underlying environmental condition. To test if a particular environmental factor $e$ tended to attract viral lineages, we approximated the following Bayes factor (BF) support[76]:

$$\mathrm{BF}_e = \left(\frac{p_e}{1-p_e}\right) \Big/ \left(\frac{0.5}{1-0.5}\right), \qquad (2)$$

where $p_e$ is the posterior probability that $E_{estimated} > E_{simulated}$, i.e., the frequency at which $E_{estimated} > E_{simulated}$ in the samples from the posterior distribution. The prior odds is 1 because we can assume an equal prior expectation for $E_{estimated}$ and $E_{simulated}$. To test if a particular environmental factor $e$ tended to repulse viral lineages, $\mathrm{BF}_e$ was approximated with $p_e$ as the posterior probability that $E_{estimated} < E_{simulated}$. These tests are similar to a previous approach using a null dispersal model based on randomisation of phylogeny branches[75].

We tested several environmental factors both as factors potentially attracting or repulsing viral lineages: elevation, main land cover variables on the study area, and climatic variables. Each environmental factor was described by a raster that defines its spatial heterogeneity (see Supplementary Table S1 for the source of each original raster file). Starting from the original categorical land cover raster with an original resolution of 0.5 arcmin (corresponding to cells ~1 km$^2$), we generated distinct land cover rasters by creating lower resolution rasters (10 arcmin) whose cell values equalled the number of occurrences of each land cover category within the 10 arcmin cells. The resolution of the other original rasters of fixed-in-time environmental factors (elevation, mean annual temperature, and annual precipitation) was also decreased to 10 arcmin for tractability, which was mostly important in the context of the second landscape phylogeographic approach detailed below. To obtain the time-series collection of temperature and precipitation rasters analysed in these first tests dedicated to the impact of environmental factors on lineage dispersal locations, we used the thin plate spline method implemented in the R package "fields" to interpolate measures obtained from the database of the US National Oceanic and Atmospheric Administration (NOAA; https://data.nodc.noaa.gov).

**Testing the impact of environmental factors on the dispersal velocity of viral lineages.** The second landscape phylogeographic testing approach aimed to test the association between several environmental factors, again described by rasters (Fig. 2), and the dispersal velocity of WNV lineages in North America. Each environmental raster was tested both as a potential conductance factor (i.e., facilitating movement) and as a resistance factor (i.e., impeding movement). In addition, for each environmental factor, we generated several distinct rasters by transforming the original raster cell values with the following formula: $v_t = 1 + k(v_o/v_{max})$, where $v_t$ and $v_o$ are the transformed and original cell values, and $v_{max}$ the maximum cell value recorded in the raster. The rescaling parameter $k$ here allows the definition and testing of different strengths of raster cell conductance or resistance, relative to the conductance/resistance of a cell with a minimum value set to "1". For each of the three environmental factors, we tested three different values for $k$ (i.e., $k = 10$, 100, and 1000).

The following analytical procedure is adapted from a previous framework[4] and can be summarised in three distinct steps. First, we used each environmental raster to compute an environmentally scaled distance for each branch in inferred and simulated trees. These distances were computed using two different path models: (i) the least-cost path model, which uses a least-cost algorithm to determine the route taken between the starting and ending points[34], and (ii) the Circuitscape path model, which uses circuit theory to accommodate uncertainty in the route taken[35]. Second, correlations between time elapsed on branches and environmentally scaled distances are estimated with the statistic $Q$ defined as the difference between two coefficients of determination: (i) the coefficient of determination obtained when branch durations are regressed against environmentally scaled distances computed on the environmental raster, and (ii) the coefficient of determination obtained when branch durations are regressed against environmentally scaled distances computed on a uniform null raster. A $Q$ statistic was estimated for each tree and we subsequently obtained two distributions of $Q$ values, one associated with inferred trees and one associated with simulated trees. An environmental factor was only considered as potentially explanatory if both its distribution of regression coefficients and its associated distribution of $Q$ values were positive[5]. Finally, the

statistical support associated with a positive $Q$ distribution (i.e., with at least 90% of positive values) was evaluated by comparing it with its corresponding null of distribution of $Q$ values based on simulated trees, and formalised by approximating a BF support using formula (2), but this time defining $p_e$ as the posterior probability that $Q_{estimated} > Q_{simulated}$, i.e., the frequency at which $Q_{estimated} > Q_{simulated}$ in the samples from the posterior distribution[33].

**Testing the impact of environmental factors on the dispersal frequency of viral lineages.** In the third landscape phylogeographic testing approach, we investigated the impact of specific environmental factors on the dispersal frequency of viral lineages: we tested if WNV lineages tended to preferentially circulate and then remain within a distinct migratory flyway. We first performed a test based on the four North American Migratory Flyways (NAMF). Based on observed bird migration routes, these four administrative flyways (Fig. 1) were defined by the US Fish and Wildlife Service (USFWS; https://www.fws.gov/birds/management/ flyways.php) to facilitate management of migratory birds and their habitats. Although biologically questionable, we here used these administrative limits to discretise the study and investigate if viral lineages tended to remain within the same flyway. In practice, we analysed if viral lineages crossed NAMF borders less frequently than expected by chance, i.e., than expected in the null dispersal model in which simulated dispersal histories were not impacted by these borders. Following a procedure introduced by Dellicour et al.[61], we computed and compared the number $N$ of changing flyway events for each pair of inferred and simulated tree. Each "inferred" $N$ value ($N_{inferred}$) was thus compared to its corresponding "simulated" value ($N_{simulated}$) by approximating a BF value using the above formula, but this time defining $p_e$ as the posterior probability that $N_{inferred} < N_{simulated}$, i.e., the frequency at which $N_{inferred} < N_{simulated}$ in the samples from the posterior distribution.

To complement the first test based on an administrative flyway delimitation, we developed and performed a second test based on flyways estimated by La Sorte et al.[36] for terrestrial bird species: the Eastern, Central and Western flyways (Supplementary Fig. S6). Contrary to the NAMF, these three flyways overlap with each other and are here defined by geo-referenced grids indicating the likelihood that studied species are migration during spring or autumn (see La Sorte et al.[36] for further details). As the spring and autumn grids are relatively similar, we built an averaged raster for each flyway. For our analysis, we then generated normalised rasters obtained by dividing each raster cell by the sum of the values assigned to the same cell in the three averaged rasters (Supplementary Fig. S6). Following a procedure similar to the first test based on NAMFs, we computed and compared the average difference $D$ defined as follows:

$$D = \sum_{i=1}^{n} \frac{v_{i,\mathrm{end}} - v_{i,\mathrm{start}}}{n}, \qquad (3)$$

where $n$ is the number of branches in the tree, $v_{i,\mathrm{start}}$ the highest cell value among the three flyway normalised rasters to be associated with the position of the starting (oldest) node of tree branch $i$, and $v_{i,\mathrm{end}}$ the cell value extracted from the same normalised raster but associated with the position of the descendant (youngest) node of the tree branch $i$. $D$ is thus a measure of the tendency of tree branches to remain within the same flyway. Each "inferred" $D$ value ($D_{inferred}$) is then compared to its corresponding "simulated" value ($D_{simulated}$) by approximating a BF value using formula (2), but this time defining $p_e$ as the posterior probability that $D_{simulated} < D_{inferred}$, i.e., the frequency at which $D_{simulated} < D_{inferred}$ in the samples from the posterior distribution.

**Testing the impact of environmental factors on the viral diversity through time.** We used the skygrid-GLM approach[9,10] implemented in BEAST 1.10.4 to measure the association between viral effective population size and four covariates: human case numbers, temperature, precipitation, and a greenness index. The monthly number of human cases were provided by the CDC and were considered with lag periods of one and two months (meaning that the viral effective population size was compared to case count data from one and two months later), as well as the absence of lag period. Preliminary skygrid-GLM analyses were used to determine from what lag period we obtained a significant association between viral effective population size and the number of human cases. We then used this lag period (of 1 month) in subsequent analyses. Data used to estimate the average temperature and precipitation time-series were obtained from the same database mentioned above and managed by the NOAA. For each successive month, meteorological stations were selected based on their geographic location. To estimate the average temperature/precipitation value for a specific month, we only considered meteorological stations included in the corresponding monthly minimum convex polygon obtained from the continuous phylogeographic inference. For a given month, the corresponding minimum convex hull polygon was simply defined around all the tree node positions occurring before or during that month. In order to take the uncertainty related to the phylogeographic inference into account, the construction of these minimum convex hull polygons was based on the 100 posterior trees used in the phylogeographic inference (see above). The rationale behind this approach was to base the analysis on covariate values averaged only over measures originating from areas already reached by the epidemic. Finally, the greenness index values were based on bimonthly Normalised Difference Vegetation Index (NDVI) raster files obtained from the NASA Earth

Observation database (NEO; https://neo.sci.gsfc.nasa.gov). To obtain the same level of precision and allow the co-analysis of NDVI data with human cases and climatic variables, we aggregated NDVI rasters by month. The visual comparison between covariate and skygrid curves shown in Fig. 5 indicates that this is an appropriate level of precision. Monthly NDVI values were then obtained by cropping the NDVI rasters with the series of minimum convex hull polygons introduced above, and then averaging the remaining raster cell values. While univariate skygrid-GLM analyses only involved one covariate at a time, the multivariate analyses included all the four covariates and used inclusion probabilities to assess their relative importance[38]. To allow their inclusion within the same multivariate analysis, the covariates were all log-transformed and standardised.

**Reporting summary**. Further information on research design is available in the Nature Research Reporting Summary linked to this article.

## Data availability

BEAST XML files of the continuous phylogeographic and skygrid-GLM analyses are available at https://github.com/sdellicour/wnv_north_america. WNV sequences analysed in the present study were available on GenBank and deposited before November 21, 2017. Accession numbers of selected genomic sequences are listed in the file "WNV_GenBank_accessions_numbers.txt" available on the GitHub repository referenced above. The source of the different raster files used in this study is provided in Supplementary Table S1. The administrative flyways were obtained from the US Fish and Wildlife Service (USFWS; https://www.fws.gov/birds/management/flyways.php).

## Code availability

The R script to run all the landscape phylogeographic testing analyses is available at https://github.com/sdellicour/wnv_north_america (https://doi.org/10.5281/zenodo.4035938).

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

## Acknowledgements
We are grateful to Frank La Sorte for sharing their estimated flyway grids. The research leading to these results has received funding from the European Research Council under the European Union's Horizon 2020 research and innovation programme (grant agreement no. 725422-ReservoirDOCS), from the Welcome Trust (Artic Network, project 206298/Z/17/Z), and from the European Union's Horizon 2020 project MOOD (grant agreement no. 874850). S.D. is supported by the *Fonds National de la Recherche Scientifique* (FNRS, Belgium) and was previously funded by the *Fonds Wetenschappelijk Onderzoek* (FWO, Belgium). S.L. and P.B. were funded by the *Fonds Wetenschappelijk Onderzoek* (FWO, Belgium). B.V. was supported by a postdoctoral grant (12U7118N) of the Research Foundation - Flanders (*Fonds voor Wetenschappelijk Onderzoek*). L.d.P. and O.G.P. are supported by the European Research Council under the European Commission Seventh Framework Programme (grant agreement no. 614725-PATHPHYLO-DYN) and by the Oxford Martin School. M.A.S. is partially supported by NSF grant DMS 1264153 and NIH grants R01 AI107034, U19 AI135995, and R56 AI149004. The content is solely the responsibility of the authors and does not necessarily represent the official views of the National Institutes of Health. P.L. acknowledges support by the Research Foundation-Flanders (*Fonds voor Wetenschappelijk Onderzoek-Vlaanderen*, G066215N, G0D5117N, and G0B9317N).

## Author contributions
S.D., K.G.A., N.D.G., O.G.P., and P.L. designed the study. S.D., M.S.G., P.B., M.A.S., and P.L. developed the analytical framework. S.D., S.L., B.V., M.S.G., P.B., K.G., N.L.M., and Y.T. analysed the data. L.d.P., A.A.F., and M.A.S. provided statistical guidance. S.D. wrote the first draft of the manuscript. All the authors interpreted and discussed the results. S.D., S.L., M.I.N., M.G., K.G.A., N.D.G., O.G.P., and P.L. discussed the epidemiological implications. All the authors edited and approved the contents of the manuscript.

## Competing interests
The authors declare no competing interests.
