## [Peer Review File · Nature Communications]

Reviewers' Comments:

Reviewer #1:

Remarks to the Author:

This manuscript by Dellicour et al. is a timely contribution that very clearly describes an approach to combining state-of-the-art phylogeographic and phylodynamic methods to formally test hypotheses about drivers of patterns of viral epidemic behaviour (i.e. dispersal location, velocity and frequency) inferred from genetic data. The authors demonstrated the utility of their approach by applying it to West Nile Virus (WNV) in North America and have identified temperature as a main predictor of viral genetic diversity through time and a substantial role for non-migratory birds and/or mosquito dispersal in spreading WNV along a longitudinal gradient.

The study design, methods and interpretation of the results are all appropriate and the study conclusions are well supported by the data presented. The work represents a significant contribution in terms of the specific findings about West Nile Virus (which will be of particular interest to those in area such as virology, ornithology and infectious disease ecology) and more importantly in terms of the combination of statistical approaches used, which will be of interest to specialists in a broader range of fields including bioinformatics, phylogenetics and infectious disease modelling.

The authors should be commended for the manuscript's clarity which presents all primary and supporting data efficiently and appropriately. In addition to the novelty and thorough application of the approach employed, the strength of the manuscript lies in the authors' careful consideration of potential limitations and confounders, and the inclusion of appropriate analyses to address these (e.g. repeat analyses with exclusion of tip nodes, restriction of analyses to specific lineages and time frames, testing of different strengths of environmental raster cell conductance / resistance, etc.).

Specific comments:

1) Page 11, Paragraph 1: Given that one of the conclusions is a substantial role for non-migratory birds, I would have liked to have seen additional comments on how this aligns with what is known from empirical data about the relative contributions of migratory versus non-migratory birds. The articles currently cited are a phylogeographic study (ref. 42) and a study that compared models of WNV movement with movement data for one migratory and one resident bird species (ref. 40). Is there relevant data regarding WNV incidence / seroprevalence in migratory versus non-migratory birds that can be discussed?

2) Page 14 Line 351: I would suggest expanding on the comment about the 1 month lag time between viral genetic diversity dynamics and human case numbers being at least partially "explained by the time needed for mosquitoes to become infectious and bite humans". If the sequences were derived primarily from mosquitoes then this could be a major part of the explanation but if a significant number of sequences were derived from humans, the entire time for transmission of virus from one human to another would come into play i.e. the time needed for the virus to become infectious in the bitten human as well as the time needed for a mosquito that has ingested an infected bloodmeal to become infectious. Depending on the extent to which sequences were derived from symptomatic cases, the period between becoming infected and showing symptoms might also have to be considered. The lag could also conceivably be due to the fact that the type of surveillance activities that produced the sequences only detected cases after the number of infections built up to a minimum threshold.

3) Page 15 line 379 states "we here propose to repeat the analysis after having discarded all the tip branches, which logically mitigates a potential impact of the sampling pattern on the outcome of this

analysis". I assume that this should have been written in the past tense as the authors described doing exactly this on page 7 (line 172) when they state " to assess the potential impact of the sampling pattern, we also repeated these tests while only considering internal tree nodes."

The same applies to lines 392 – 395 which also present the method used in the future tense (as proposed work). This should be revised to read "To mitigate spatial heterogeneity as much as possible, we used the continuous phylogeographic reconstruction to define successive minimum convex hull polygons delimitating the study area at each time slice. These polygons were used to extract environmental values that were then averaged to obtain a single environmental value per time slice considered in the skygrid-GLM analysis."

4) Page 16 Lines 423 to 425: Were sequence dates considered when doing this subsampling? Or were the subsampled sequences unique in terms of both location and year of origin? Please clarify.

5) Page 18 Line 481: Change "On the opposite..." to " "On the other hand..."

6) Page 19 Line 503 – "...consisted in testing" should be amended to "consisted of testing"

7) Figure 5 – Where is the significance of the asterisk next to the axis label "number of cases" explained?

Reviewer #2:

Remarks to the Author:

The manuscript by Delicour et al. describes computational analyses implemented to better understand the velocity and pattern of West Nile virus dispersal and diversification in the U.S. In addition, the role of various environmental factors in driving WNV phylogeography and phylodynamics is considered in an analytical framework. WNV provides the ideal system for such an approach given that it is/was a novel pathogen invading a naïve host environment with a relatively complete genetic and epidemiological record. The manuscript is very well written and organized. The figures and methodology are clear and well-presented. While the novelty of the analyses seem only incremental in nature, they are appropriate and could be broadly utilized for other systems if extensive data were available. The most significant concern/consideration with this and any similar study is the appropriateness of the scale utilized and the role of sampling bias in driving results. The authors are limited to what is available and they address these concerns to some extent, but some additional caution with interpretation in the context of what is known about WNV evolution and ecology is warranted. Specific concerns/comments are listed in their order of appearance below.

Line 105- 801 viral genomes. The number of isolates is a strength relative to past studies but the authors never disclose the geographic coding of the strains proportional to the total number. You can basically see the geographic distribution in the figures but having this information is critical to assess the extent and influence of sampling biases.

Line 107- Frequent long-distance dispersal events. It would seem this is one place where sampling bias could play a large role. i.e. A WNV strain could appear to jump from NY to Chicago area (both highly sampled) simply because there is little sampling in between.

Line 121- 'Five subsets of WNV lineages'. What defines a subset and a lineage? A well-supported clade? Are there numerous shared mutations within these clades? Additionally, the use of the term lineages here and throughout is problematic given the historical separation of WNV into distinct

lineages that are previously defined. All of the strains analyzed here are lineage 1A strains. Perhaps change to clades/clusters/branches or something similar?

Line 136- Point here and elsewhere regarding higher dispersal velocity of SW03 genotype. Could it not be that mutations associated with SW03 have been independently selected numerous times and that this phenomenon could be perceived as higher dispersal velocity? Also, SW03 and WN02 are discussed as if they are independent genotypes. SW03 is in fact a WN02 genotype strain, so they should not be considered discretely, but continuously. Perhaps it's more accurate to conclude that the secondary mutations associated with SW03 accelerated the dispersal of WN02.

Line 146- Monthly time series factors. Was this the month preceding or following the isolation? The scale of these environmental measurements (both in time and space) would clearly have a significant impact on the result. Different scales could be justified and the authors are limited to the data that is available, so what they did is likely OK, yet it would be good to discuss a bit more what is already known about the relationship between WNV and temperature, precipitation and land in the context of these analyses/results.

Line 173- Internal tree nodes. Here the authors are acknowledging and attempting to correct for sampling bias but they don't explain what that bias could be. In addition, this analysis is useful and should be retained but the limitations should be acknowledged. If the tips were different the inferred internal nodes would also be different, so bias is not eliminated here.

Line 200- Temperature as a conductance factor. The novel analyses that result in this finding are commendable, but the question remains if it is simply that there was more sampling when temperature was higher. Studies looking at local circulation of WNV have found a lot of diversity on small temporal and geographic scales and it seems anytime there is more sampling studies have found more diversification than would have been predicted by previous studies looking at broader samples. There is substantial evidence that increased temperature is associated with increased WNV activity (which should be more completely acknowledged in the discussion) so the finding is likely valid but I wonder if temporal bias is removed (i.e. the same analysis is done using 20 samples from each year) if the finding would be the same? This also applies to the finding that more human cases equate to more genetic diversity.

Line 311- It is not clear what is meant by 'environmental adaptation'. Also, it should be noted in the paper that studies assessing the phenotypic impact of the mutations associated with SW03 have not found that they confer any fitness advantage in host or vector.

Line 319-332- It's confusing to the reader that you could simultaneously conclude that flyways do not contribute to clustering but also it seems cannot reject that they do? Here, again, it might be useful to add some WNV biology/ecology. High levels of WNV, and subsequently diversification/spread are not generally occurring when birds are migrating, so it would make sense that while flyways could contribute occasionally to long-distance, seasonal dispersal, they would not be the primary driver of diversification.

Line 351- The lag between infection and diagnosis is well established. In addition to the extrinsic incubation period in mosquitoes (as stated), it is a result of the incubation period in humans, the time between symptom onset and diagnosis, and the time between diagnosis and reporting.

Reviewer 1

This manuscript by Dellicour et al. is a timely contribution that very clearly describes an approach to combining state-of-the-art phylogeographic and phylodynamic methods to formally test hypotheses about drivers of patterns of viral epidemic behaviour (i.e. dispersal location, velocity and frequency) inferred from genetic data. The authors demonstrated the utility of their approach by applying it to West Nile Virus (WNV) in North America and have identified temperature as a main predictor of viral genetic diversity through time and a substantial role for non-migratory birds and/or mosquito dispersal in spreading WNV along a longitudinal gradient.

The study design, methods and interpretation of the results are all appropriate and the study conclusions are well supported by the data presented. The work represents a significant contribution in terms of the specific findings about West Nile Virus (which will be of particular interest to those in area such as virology, ornithology and infectious disease ecology) and more importantly in terms of the combination of statistical approaches used, which will be of interest to specialists in a broader range of fields including bioinformatics, phylogenetics and infectious disease modelling.

The authors should be commended for the manuscript's clarity which presents all primary and supporting data efficiently and appropriately. In addition to the novelty and thorough application of the approach employed, the strength of the manuscript lies in the authors' careful consideration of potential limitations and confounders, and the inclusion of appropriate analyses to address these (e.g. repeat analyses with exclusion of tip nodes, restriction of analyses to specific lineages and time frames, testing of different strengths of environmental raster cell conductance / resistance, etc.).

Answer: Thank you very much for the positive feedback.

Specific comments:

1) Page 11, Paragraph 1: Given that one of the conclusions is a substantial role for non-migratory birds, I would have liked to have seen additional comments on how this aligns with what is known from empirical data about the relative contributions of migratory versus non-migratory birds. The articles currently cited are a phylogeographic study (ref. 42) and a study that compared models of WNV movement with movement data for one migratory and one resident bird species (ref. 40). Is there relevant data regarding WNV incidence / seroprevalence in migratory versus non-migratory birds that can be discussed?

Answer: The Reviewer makes an important point about the lack of empirical data linking WNV movement to migratory or resident birds. While there is literature on WNV seroprevalence in birds (e.g. Dusek *et al.* 2009, PMID: 19996451; Loss *et al.* 2008, PMID: 19034529), the studies are not large enough to quantify the relative difference in infection rates among the groups of birds. This is one of the fundamental knowledge gaps in WNV research, due in part by the 100s of bird species that are relevant for transmission and the massive studies that would need to be designed to capture these data – which is no easy feat for some large migratory bird species. A large systematic review of the literature could help to reveal some trends, but it would still likely be underpowered and biased towards easy-to-capture birds. Anything that we can add to the manuscript from these studies would be speculation or too general to be helpful (e.g. yes, both migratory and residential birds can become infected, and the rates significantly change among species and years). Unfortunately, our best evidence to date comparing the bird groups are from phylogenetic and modelling studies.

2) Page 14 Line 351: I would suggest expanding on the comment about the 1 month lag time between viral genetic diversity dynamics and human case numbers being at least partially “explained by the time needed for mosquitoes to become infectious and bite humans”. If the sequences were derived primarily from mosquitoes then this could be a major part of the explanation but if a significant number of sequences were derived from humans, the entire time for transmission of virus from one human to another would come into play i.e. the time needed for the virus to become infectious in the bitten human as well as the time needed for a mosquito that has ingested an infected bloodmeal to become infectious. Depending on the extent to which sequences were derived from symptomatic cases, the period between becoming infected and showing symptoms might also have to be considered. The lag could also conceivably be due to the fact that the type of surveillance activities that produced the sequences only detected cases after the number of infections built up to a minimum threshold.

Answer: We thank the Reviewer for this suggestion. As now explicitly specified in the Methods section, most sequences were derived from mosquitoes (~50%) and birds (~44%), with very few (~5%) from humans. Viremic humans are rarely detected and thus are not rich sources for sequencing. Furthermore, humans are dead-end hosts, so the virus cannot be passed from human to human with a mosquito intermediate. In addition, human case counts are in theory backdated to the date of onset of illness, meaning that incubation time in humans (and/or the time needed to report the symptoms) should not have contributed to this 1 month lag. To clarify this aspect, we have now extended that part of the discussion as follows:

“Such lag could, at least in part, be explained by the time needed for mosquitos to become infectious and bite humans. Because human case counts are in theory backdated to the date of onset of illness, incubation time in humans should not contribute to this lag”.

3) Page 15 line 379 states "we here propose to repeat the analysis after having discarded all the tip branches, which logically mitigates a potential impact of the sampling pattern on the outcome of this analysis". I assume that this should have been written in the past tense as the authors described doing exactly this on page 7 (line 172) when they state "to assess the potential impact of the sampling pattern, we also repeated these tests while only considering internal tree nodes."

Answer: Indeed, we have modified the sentence accordingly.

The same applies to lines 392 – 395 which also present the method used in the future tense (as proposed work). This should be revised to read “To mitigate spatial heterogeneity as much as possible, we used the continuous phylogeographic reconstruction to define successive minimum convex hull polygons delimitating the study area at each time slice. These polygons were used to extract environmental values that were then averaged to obtain a single environmental value per time slice considered in the skygrid-GLM analysis.”

Answer: Corrected.

4) Page 16 Lines 423 to 425: Were sequence dates considered when doing this subsampling? Or were the subsampled sequences unique in terms of both location and year of origin? Please clarify.

Answer: No, the sequence dates are not considered in subsampling procedure. The considered sentence finalised the description of the subsampling by phylogenetic clusters, the description of which starts in the previous sentences. To maintain the connection between these sentences, we now use the expression “phylogenetic clusters” (instead of simply “cluster”).

5) Page 18 Line 481: Change “On the opposite...” to “On the other hand...”

Answer: Corrected.

6) Page 19 Line 503 – “...consisted in testing” should be amended to “consisted of testing”

Answer: Corrected.

7) Figure 5 – Where is the significance of the asterisk next to the axis label “number of cases” explained?

Answer: We apologise for the confusion, that asterisk is not necessary and has now been removed.

Reviewer 2

The manuscript by Dellicour et al. describes computational analyses implemented to better understand the velocity and pattern of West Nile virus dispersal and diversification in the U.S. In addition, the role of various environmental factors in driving WNV phylogeography and phylodynamics is considered in an analytical framework. WNV provides the ideal system for such an approach given that it is/was a novel pathogen invading a naïve host environment with a relatively complete genetic and epidemiological record. The manuscript is very well written and organized. The figures and methodology are clear and well-presented. While the novelty of the analyses seem only incremental in nature, they are appropriate and could be broadly utilized for other systems if extensive data were available. The most significant concern/consideration with this and any similar study is the appropriateness of the scale utilized and the

role of sampling bias in driving results. The authors are limited to what is available and they address these concerns to some extent, but some additional caution with interpretation in the context of what is known about WNV evolution and ecology is warranted. Specific concerns/comments are listed in their order of appearance below.

Answer: Thank you for this positive feedback on our work.

The first two maps reported in the new version of Figure S1: distribution of human cases of West Nile fever and sampling maps of WNV genomes analysed in the present study. We first report the \log_{10} -transformed number of reported human cases per US county (source: Centers for Disease Control and Prevention, www.cdc.gov/westnile). In the subsequent sampling map, we report the sample positions coloured according to sampling times.

Line 105- 801 viral genomes. The number of isolates is a strength relative to past studies but the authors never disclose the geographic coding of the strains proportional to the total number. You can basically see the geographic distribution in the figures but having this information is critical to assess the extent and influence of sampling biases.

Answer: We agree and have now generated a new map integrated into Figure S1 (see above) that displays the spatial distribution of confirmed human cases of West Nile fever. In the absence of similar data available for infected mosquito and/or bird species, this new map allows for a direct visual comparison between the distribution of human cases and the sampling of viral genomes considered in our study. It highlights a good correspondence between West Nile fever prevalence in human population and sampling density in most areas associated with the highest numbers of reported cases (e.g. Los Angeles, Houston, Dallas, Chicago, New York), but also some under-sampled locations (e.g. in Colorado). We also refer to our other answers to subsequent comments related to the sampling effort aspect.

New Figure S4: Comparison between the maximum clade credibility (MCC) tree branch durations and geographic distances travelled by these MCC tree branches. Each dot corresponds to a distinct MCC tree branch and is coloured according to the time of occurrence of the youngest node of that branch. We also report the coefficient of determination (R^2) obtained from the linear regression between branch durations and geographic distances. This scatterplot highlights the occurrence of both relatively slow and relatively fast long-distance dispersal events associated with MCC tree branches, which is also reflected by the low correlation estimated between branch durations and geographic distances travelled by those branches (coefficient of determination $R^2 = \sim 5\%$).

Line 107- Frequent long-distance dispersal events. It would seem this is one place where sampling bias could play a large role. i.e. a WNV strain could appear to jump from NY to Chicago area (both highly sampled) simply because there is little sampling in between.

Answer: We thank the Reviewer for pointing out this important aspect. Because of the sampling pattern, the detection of a long-distance dispersal event associated with a given phylogenetic branch could indeed result from a long undersampled transmission chain that gradually crossed the distance, i.e. transiting

through diverse locations not necessarily in between the two nodes of that branch. In that case however, this should be reflected by a long time span inferred for the phylogenetic branch under consideration. Reporting long-distance dispersal events without analysing their associated dispersal velocity was indeed a shortcoming in the previous version of our manuscript: a phylogenetic branch connecting two distant locations does not necessarily correspond to a fast long-distance dispersal event, hence the importance of also investigating its dispersal velocity. To address this issue, we have now added a comparison between the maximum clade credibility (MCC) tree branch durations and geographic distances travelled by these MCC tree branches (new Figure S5, see above). This new analysis highlights the occurrence of both relatively slow and relatively fast long-distance dispersal events associated with MCC tree branches, which is also reflected by the low correlation estimated between branch durations and geographic distances travelled by those branches (coefficient of determination $R^2 = \sim 5\%$). As also reported by the snapshots of WNV phylogenetic branches dispersal history in Figure S3, our analysis demonstrates that some of these long-distance dispersal events were notably fast, with >2000 km travelled in only a couple of months. This is now explicitly reported in the text.

Line 121- ‘Five subsets of WNV lineages’. What defines a subset and a lineage? A well-supported clade? Are there numerous shared mutations within these clades? Additionally, the use of the term lineages here and throughout is problematic given the historical separation of WNV into distinct lineages that are previously defined. All of the strains analyzed here are lineage 1A strains. Perhaps change to clades/clusters/branches or something similar?

Answer: We agree with the Reviewer that the word “lineage” can be used to designate a clade or a strain, but it can also be used to designate phylogenetic branches. As we used expressions like “lineage dispersal history/dynamics/velocity” in previous related works (e.g. PMID: 28651357, PMID: 31535448, PMID: 31790143), we would prefer to keep this terminology. However, we agree that this particular sentence was confusing, and we have now rewritten it as follows:

“We also compared the dispersal velocity estimated for five subsets of WNV phylogenetic branches (Fig. 3): branches occurring during (before 2002) and after the expansion phase (after 2002), as well as branches assigned to each of the three commonly defined WNV genotypes that circulated in North America (NY99, WN02, and SW03; Figs. S1-S2).”

Line 136- Point here and elsewhere regarding higher dispersal velocity of SW03 genotype. Could it not be that mutations associated with SW03 have been independently selected numerous times and that this phenomenon could be perceived as higher dispersal velocity?

Answer: The tree with tip nodes coloured according to WNV genotypes (Figure S2) clearly tends to discard the hypothesis of multiple independent mutations leading to the transition from genotype WN02 to SW03. We are not saying that it is certain that there was only one mutation event, but that it is highly unlikely that there were such multiple mutation events.

Also, SW03 and WN02 are discussed as if they are independent genotypes. SW03 is in fact a WN02 genotype strain, so they should not be considered discretely, but continuously. Perhaps it’s more accurate to conclude that the secondary mutations associated with SW03 accelerated the dispersal of WN02.

Answer: As SW03 genotypes constitute a distinct monophyletic clade, we would prefer to maintain the current terminology regarding the distinction between these two genotypes. This terminology is also frequently used in the literature, as well as on the platform Nextstrain (<https://nextstrain.org/WNV>).

Line 146- Monthly time series factors. Was this the month preceding or following the isolation? The scale of these environmental measurements (both in time and space) would clearly have a significant impact on the result. Different scales could be justified and the authors are limited to the data that is available, so what they did is likely OK, yet it would be good to discuss a bit more what is already known about the relationship between WNV and temperature, precipitation and land in the context of these analyses/results.

Answer: Time series environmental data (temperature, precipitation) were obtained for each month and were not lagged compared to sampling dates. We use a time interval of one month as common time interval for all covariates tested in our skygrid-GLM analyses. Indeed, except for the greenness index (NDVI) for which we initially obtained bimonthly data, we managed to gather monthly data for the other covariates. This aspect is now explicitly stated in the related Methods section. We acknowledge that the length of the time interval could potentially have an impact on the outcome of these analyses. However,

the visual comparison between covariate and skygrid curves shown in Figure 5 indicates that this an appropriate level of precision.

Line 173- Internal tree nodes. Here the authors are acknowledging and attempting to correct for sampling bias but they don't explain what that bias could be.

Answer: We perform this additional test in the specific context of the analysis that aims at investigating the impact of environmental conditions on the dispersal locations of WNV phylogenetic branches. Indeed, as stated in the text, "in this test, half of the environmental values will be extracted at tip node locations, which are directly determined by the sampling effort". In other words, because the sampling locations directly determine half of the environmental values considered in that analysis, we aimed to perform this robustness test only focusing on internal nodes (but see our related answer just below). In addition, as the full transmission chain is unknown, and given the good correspondence we now demonstrated between West Nile fever prevalence and sampling density in most areas associated with the highest numbers of reported cases, we prefer to talk about "heterogeneous sampling effort" than "sampling bias".

In addition, this analysis is useful and should be retained but the limitations should be acknowledged. If the tips were different the inferred internal nodes would also be different, so bias is not eliminated here.

Answer: We completely agree with the Reviewer that, since internal nodes are phylogenetically-connected to tip nodes, this additional test does not *fully* remove the potential impact of the sampling effort on the outcome of the analysis. We do not consider this approach as an attempt to remove the impact of heterogeneous sampling effort, but to assess how sensitive our results are to this. If the impact would be strong, we believe this should already be noticeable in the comparison with and without tip branches. To acknowledge this aspect, we have edited the corresponding text as follows:

"Because these tests are directly based on the environmental values extracted at internal and tip node positions, their outcome can particularly be impacted by the nature of sampling. Indeed, half of the node positions, i.e. the tip node positions, are directly determined by the sampling. To assess the sensitivity of the tests to heterogeneous sampling, we also repeated these tests while only considering internal tree nodes. Since internal nodes are phylogeographically-linked to tip nodes, discarding tip branches can only mitigate the direct impact of the sampling pattern on the outcome of the analysis."

Line 200- Temperature as a conductance factor. The novel analyses that result in this finding are commendable, but the question remains if it is simply that there was more sampling when temperature was higher.

Answer: This analysis investigates to what extent phylogenetic branches that traverse areas with lower/higher temperatures are associated with lower/higher dispersal velocity. Even when areas with high and low temperatures would be disproportionally represented, the number of branches in those regions is not what informs the test, but it is the dispersal velocity of those branches. Branches in these areas need to have a different average velocity for the test to be significant. So, one could argue that sampling heterogeneity may actually reduce the power of the test. This aspect is discussed in the text:

"While the resolution of phylogeographic analyses will always depend on the spatial granularity of available samples, they can still be powerful in elucidating the dispersal history of sampled lineages. When testing the impact of environmental factors on lineage dispersal velocity and frequency, heterogeneous sampling density will primarily affect statistical power in detecting the impact of relevant environmental factors in under- or unsampled areas (Dellicour *et al.* 2017)."

Studies looking at local circulation of WNV have found a lot of diversity on small temporal and geographic scales and it seems anytime there is more sampling studies have found more diversification than would have been predicted by previous studies looking at broader samples. There is substantial evidence that increased temperature is associated with increased WNV activity (which should be more completely acknowledged in the discussion) so the finding is likely valid but I wonder if temporal bias is removed (i.e. the same analysis is done using 20 samples from each year) if the finding would be the same? This also applies to the finding that more human cases equate to more genetic diversity.

Answer: This analysis is not based on a time-series variable but on averaged values (annual mean temperature values). Therefore, subsampling by year would not be meaningful in this specific situation. In lines 355-373, we include a very thorough discussion on the role of temperature on WNV transmission (which includes 8 references), ranging from its impact on mosquito reproduction, development, behaviours, and virus incubation/replication. Temperature is a widely-accepted risk factor for WNV

transmission. Finally, since the majority of sequences were not derived from humans, finding human cases is not directly linked to more virus genetic diversity (though bird and mosquito infections are).

Line 311- It is not clear what is meant by ‘environmental adaptation’. Also, it should be noted in the paper that studies assessing the phenotypic impact of the mutations associated with SW03 have not found that they confer any fitness advantage in host or vector.

Answer: We apologise for the previous lack of clarity. “Environmental adaptations” was a reference to the environmental conditions mentioned in the previous sentence. We have now clarified that connection as follows:

“we found that the dispersal of SW03 genotype is faster than WN02 and also preferentially in shrublands and at higher temperatures. At face value, it appears that the substitutions that define the SW03 genotype, NS4A-A85T and NS5-K314R45, may be signatures of adaptations to such specific environmental conditions.”

Furthermore, we have now also explicitly acknowledged the lack of evidence that substitutions associated with SW03 confer any fitness advantage in host or vector:

“It is also important to note that to date, no specific phenotypic advantage has been observed for SW03 genotypes compared to WN02 genotypes (Worwa *et al.* 2018, Duggal *et al.* 2019).”

Line 319-332- It’s confusing to the reader that you could simultaneously conclude that flyways do not contribute to clustering but also it seems cannot reject that they do? Here, again, it might be useful to add some WNV biology/ecology. High levels of WNV, and subsequently diversification/spread are not generally occurring when birds are migrating, so it would make sense that while flyways could contribute occasionally to long-distance, seasonal dispersal, they would not be the primary driver of diversification.

Answer: The Reviewer’s comment made us realise that there was a mistake in the relevant sentence, which may have been the cause of the Reviewer’s confusion – we have now restated this as follows:

“Our results are, however, not in contradiction with the already established role of migratory birds in spreading the virus (Reed *et al.* 2003, Dusek *et al.* 2009)14/08/2020 16:11:00, but we do not find evidence that viral lineage dispersal is structured by flyway. Specifically, our test does not reject the null hypothesis of absence of clustering by flyways, which at least signals that the tested flyways do not have a discernible impact on WNV lineages circulation.”

The impact of migratory bird in WNV maintenance and spatial spread is yet to be fully explored. Our results hint that they did not have a statistically discernible impact on the clustering of WNV lineages circulation. Our work cannot however exclude that they, occasionally, participate to WNV transmission and spread. Several scenarios, taken alone or in combination, might explain this observation: e.g. decoupling between migration and peak mosquito season (such as suggested by the Reviewer), heterogeneity in bird migration spatio-temporal pattern (especially if only a few species are involved), or marginal involvement of migratory birds... We believe that at this point they all are highly speculative, only new empirical data could help favour one or the other. An additional sentence has been added to the manuscript:

“Dissecting the precise involvement of migratory bird in WNV spread thus require additional collection of empirical data.”

Line 351- The lag between infection and diagnosis is well established. In addition to the extrinsic incubation period in mosquitoes (as stated), it is a result of the incubation period in humans, the time between symptom onset and diagnosis, and the time between diagnosis and reporting.

Answer: Thank you for pointing that aspect – we have now addressed this in our reply to a related comment by Reviewer #1 (specific comment #2).

Reviewers' Comments:

Reviewer #1:

Remarks to the Author:

The authors have satisfactorily addressed all of my concerns in this revision.

Reviewer #2:

Remarks to the Author:

The authors have thoroughly and adequately responded to all critiques. The revised manuscript is suitable for publication.